# Phenolic Compound Profiles, Cytotoxic, Antioxidant, Antimicrobial Potentials and Molecular Docking Studies of *Astragalus gymnolobus* Methanolic Extracts

**DOI:** 10.3390/plants13050658

**Published:** 2024-02-27

**Authors:** Esra Aydemir, Elif Odabaş Köse, Mustafa Yavuz, A. Cansu Kilit, Alaaddin Korkut, Serap Özkaya Gül, Cengiz Sarikurkcu, Mehmet Engin Celep, R. Süleyman Göktürk

**Affiliations:** 1Department of Biology, Faculty of Science, Akdeniz University, Antalya TR-07058, Turkey; myavuz@akdeniz.edu.tr (M.Y.); cansukilit@akdeniz.edu.tr (A.C.K.); 202151005003@ogr.akdeniz.edu.tr (A.K.); 202151006001@ogr.akdeniz.edu.tr (S.Ö.G.); gokturk@akdeniz.edu.tr (R.S.G.); 2Medical Laboratory Program, Vocational School of Health Services, Akdeniz University, Antalya TR-07058, Turkey; elifkose@akdeniz.edu.tr; 3Department of Analytical Chemistry, Faculty of Pharmacy, Afyonkarahisar Health Sciences University, Afyonkarahisar TR-03100, Turkey; cengiz.sarikurkcu@afsu.edu.tr; 4Department of Pharmacognosy, Faculty of Pharmacy, Yeditepe University, Atasehir, Istanbul TR-34755, Turkey; ecelep@yeditepe.edu.tr

**Keywords:** apoptosis, *Astragalus*, antimicrobial, antioxidant, cytotoxicity

## Abstract

Since *Astragalus* is a genus with many important medicinal plant species, the present work aimed to investigate the phytochemical composition and some biological activities of *Astragalus gymnolobus*. The methanolic fractions of four organs (stems, flowers, leaves, root and whole plant) were quantified and identified by Liquid Chromatography Electrospray Ionization Tandem Mass Spectrometry (LC–ESI–MS/MS) analysis. Hesperidin, hyperoside, *p*-hydroxybenzoic acid, protocatechuic acid and *p*-coumaric acid were identified as main compounds among the extracts. Among all cells, leaf methanol (Lm) extract had the highest cytotoxic effect on HeLa cells (IC_50_ = 0.069 μg/mL). Hesperidin, the most abundant compound in *A. gymnolobus* extract, was found to show a strong negative correlation with the cytotoxic effect observed in HeLa cells according to Pearson correlation test results and to have the best binding affinity to targeted proteins by docking studies. The antimicrobial activity results indicated that the most susceptible bacterium against all extracts was identified as *Streptococcus pyogenes* with 9–11 mm inhibition zone and 8192 mg/mL MIC value. As a result of the research, it was suggested that *A. gymnolobus* could be considered as a promising source that contributes to the fight against cancer.

## 1. Introduction

Although every day a new method for treatment is developed, cancer is still the most important health problem worldwide. The current treatments include chemotherapy, radiotherapy and surgical interventions. Methods used for cancer treatment, especially chemotherapy and radiotherapy, cause toxic side effects on patients. Therefore, the success expected from the treatment cannot be fully achieved and, moreover, a troublesome period after treatment for the patient occurs. To overcome these problems, there is a focus on using and/ or creating alternative treatments and therapies which should be more selective against cancer cells and cause fewer toxic responses in normal cells. Scientific research area is drawing its attention towards naturally-derived compounds since the medicinal plants have been widely used for thousands of years in folk medicines to treat different diseases. Many plants are known to contain compounds with anticancer properties [1].

The genus *Astragalus* L., which belongs to the Papilionoideae subfamily of Fabaceae (Leguminosae) family, is represented by approximately 2900 species are widely distribut-ed throughout the temperate and arid regions principally in Europe, Asia, and North America [2]. There are 478 taxa in Turkey mostly growing in steppes and high mountains and 202 of them are endemic [3]. *Astragalus* species have high nutritional values and many species are used as fuel and animal feed as well [4]. In spite of widespread distribution of *Astragalus* species in Turkey only a few species have gained economic importance in the production of gum tragacanth, a renowned nutrient and also as glue in processing of cosmetics and pharmaceuticals i.e., emulsions and suspensions [5]. In addition, there are several ethnobotanical field records concerning the use of some *Astragalus* species in the treatment of leukemia and also for wound healing in Turkish folk medicine [6,7]. Among the worldwide members of *Astragalus*, *A. membranaceus* Moench is the most popular species in Traditional Chinese Medicine. The roots of the plant have been particularly prescribed to strengthen and regulate the immune system as supplement therapy for cancers.

Saponins, flavonoids, and polysaccharides are believed to be the principle active constituents of *Astragalus*. It also includes components such as anthraquinones, alkaloids, amino acids, β-sitosterol, and metallic elements [8]. In particular, alkaloids, saponins, flavonoids, and anthraquinones are thought to be the main compounds responsible for their potential biological effects of *Astragalus* species [9]. In this context, *Astragalus*-derived extracts and isolated components have been shown to exhibit a wide range of biological properties, such as anti-inflammatory [10,11,12], immunostimulant [6,7,13], anti-protozoal [14], antibacterial, hepatoprotective, cardio-protective [10], antiviral [10,15], wound healing [16], expectorant, analgesic, sedative, hypotensive [17,18] and cytotoxic activities [19].

*Astragalus gymnolobus* Fisch is an endemic species of rattleweed that is unique to our country and grows steppes, fields, forest areas and slopes between 1000–1800 m. This species which blooms in May and June is distributed in Central Anatolia, Southeast and Eastern Anatolia and Mediterranean Regions. The flowers are usually white, cream or pale yellow. Its legumen type fruit is ovoid, oblong and naked. There are studies in the literature that investigate phytochemical analysis and the biological characteristics of *A. gymnolobus* [9,20,21]. The main flavonoids analyzed in these studies were hesperidin, hyperoside and rutin, which were also reported as compounds responsible for biological activity [9,20]. In light of this information, we aimed to determine identification and quantification of phenolic compounds in methanol extracts obtained from different regional organs (stems, flowers, leaves, root, whole plant) of *A. gymnolobus* and to further elucidate the similarities and differences in their chemical components using LC–ESI–MS/MS. Besides the antioxidant, antimicrobial and anticancer potentials of all these extracts will be investigated in this study. Pearson correlation analysis will be performed to establish a statistical correlation between the phytochemical compositions of the extracts and their anticancer activity potential. Finally, according to the information obtained from the biological activity results, docking studies will be carried out as an important step towards understanding the interaction mode of the most abundant compound in the extract as a result of phytochemical analysis. Based on our knowledge of the literature, no comprehensive study has been found that defines the contents and investigates the biological activities of methanol extracts obtained from each different parts of *A. gymnolobus* naturally growing in Antalya, Turkey.

## 2. Results

### 2.1. Extraction of Plants

The plant materials, i.e., whole plant and the dissected plant parts (root, stem, leaf and flower parts) were extracted with methanol, then evaporated to dryness under reduced pressure in the rotary evaporator. The yields of the methanolic extracts of root (Rm), stem (Sm), flower (Fm), leaf (Lm) and whole plant (WPm) of *A. gymnolobus*, were calculated as 14.33%, 11.95%, 7.84%, 24.38% and 19.27% (*w*/*w*), respectively.

### 2.2. Phytochemical Analysis of Extracts

The amounts of thirty different standard phytochemicals in methanol extracts from different parts of *A. gymnolobus* were detected using previously validated Liquid Chromatography Electrospray Ionization Tandem Mass Spectrometry (LC–ESI–MS/MS) method and the results are given in Table 1. Hesperidin, hyperoside, *p*-hydroxybenzoic acid, protocatechuic acid and *p*-coumaric acid were identified as main compounds in the extracts. *A. gymnolobus* Rm and Sm extracts had higher hesperidin and hyperoside contents than those the other parts as well as (+)-catechin content. In addition, the presence of pyrocatechol, protocatechuic acid, caffeic acid, verbascoside, taxifolin, sinapic acid, luteolin 7-glucoside, rosmarinic acid, *o*-coumaric acid, pinoresinol, eriodictyol, luteolin and apigenin was not detected in any part of *A. gymnolobus*.

### 2.3. Cytotoxic Effects of A. gymnolobus Extracts

Cell viability of *A. gymnolobus* Fm, Lm, Sm, Rm and WPm extracts were evaluated for all cells after 24, 48 and 72 h of incubation, and the extracts with IC_50_ values are shown in Table 2 together with their incubation times. The OD values obtained from WST-1 test were evaluated using Graph-Pad Prism, version 4.00 (Graph-Pad Software, San Diego, CA, USA), graphed separately for each cell line with SigmaPlot 10 and presented in Appendix A.

### 2.4. Colorimetric Protease (Caspase-2, -3, -6, -8, -9) Assay

Whether the extracts at IC_50_ concentrations for each incubation period, caused changes in some important caspase activities involved in apoptosis mechanism in cells were investigated by using colorimetric protease kit. Amino acid sequences unique to each caspase; VDVAD (Caspase-2), DEVD (Caspase-3), VEID (Caspase-6), IETD (Caspase-8) and LEHD (Caspase-9) were used. The absorbance values, obtained as a result of the release of the pNA portion by cleavage of the peptides labeled with *p*-nitroaniline in the kit by caspases, were determined at 405 nm. The absorbance values obtained for each extract were compared with the control group and the changes caused by the extracts on the caspase activity in the cells were calculated. The fold increase in caspase activities caused by the exracts at IC_50_ concentrations for each incubation period for A549, HeLa and MDA-MB-231 cells are shown in Figure 1.

### 2.5. Cellular DNA Fragmentation

ELISA kit was used to determine whether the extracts trigger the formation of DNA fragmentation in cells at the determined IC_50_ concentrations. The BrdU included in the kit is a non-radioactive thymidine analog and binds to genomic DNA to determine whether the DNA is fragmented. In order to determine cellular DNA fragmentation; The cells were treated with the extracts prepared in ¼IC_50_, ½IC_50_, IC_50_, 2xIC_50_, 4xIC_50_ concentrations for the WST-1-determined incubations. The absorbances of BrdU-labeled DNA fragments were measured at a wavelength of 450 nm. Apoptotic DNA fragmentation is directly proportional to the increased absorbance. Among the cells, the DNA fragmentations of the most cytotoxic extracts are shown in Figure 2 for A549, HeLa and MDA-MB-231 cells.

### 2.6. Antimicrobial Activity

In this study, the methanol extracts of *A. gymnolobus* were screened for antibacterial activity against 16 bacterial strains according to disc diffusion and broth microdilution methods. The results of disc diffusion test on methanol extracts from different parts of *A. gymnolobus* are given in Table 3. According to the test results, all of the extracts showed antibacterial effect against only *S. pyogenes* with a zone diameter of 9–11 mm. All three *Staphylococcus aureus* strains showed sensitivity to Fm, and WPm extracts with zone diameters of 8–9 mm. The most susceptible bacterium against all extracts was identified as *S. pyogenes*. The WPm extract showed the highest activity against these bacteria. The extracts showed no activity against other bacteria. When the broth microdilution test results were evaluated, it was determined that the tested extracts were effective only against *S. pyogenes* with 8192 µg/mL MIC value. MIC values against all other tested bacteria were determined as >8192 µg/mL.

### 2.7. Determination of Antioxidant Activities of Extracts

In this study, *A. gymnolobus* methanolic extracts were investigated for their antioxidant activity with nine different assays. Results of the total phenolic and flavonoid content, total antioxidant capacity, and antioxidant capacity equivalent to Trolox were reported in Table 4. The results of DPPH radical scavenging effects, Iron reducing antioxidant power, Copper ion reducing effect (CUPRAC) and inhibiting lipid peroxidation in β-carotene/linoleic acid system were reported in Table 5. Unfortunately, superoxide radical scavenging effect was not detected in any of the extracts.

### 2.8. Pearson Correlation Analysis

When compared with other components in the *A. gymnolobus* methanol extract, the most common substances in terms of quantity are respectively; hesperidin, hyperoside, *p*-hydroxybenzoic acid and (+)-Catechin. The correlation between the changes in the amounts of the compounds obtained in the content analysis and the IC_50_ values determined because of the cell viability analysis was examined by Pearson analysis. In the pearson correlation obtained, gallic acid (r = 0.99, *p* = 0.02) in 48 h incubation period, syringic acid (r = 0.98, *p* = 0.02) in 72 h incubation period in MDA-MB-231 cells, ferulic acid in 24 h and 48 h incubation in HeLa cells (r = 0.99 *p* = 0.04), showed positive correlation with apigenin 7-glucoside (r = 0.99 *p* = 0.05) during the 24 h incubation period. 3-hydroxybenzoic acid (r = −0.98 *p* = 0.04), apigenin 7-glucoside (r = −0.99 *p* = 0.04) in 48 h incubation in MDA-MB-231 cells, catechin (r = −0.99 *p* = 0.008), 3- hydroxybenzoic acid (r = −0.98 *p* = 0.01) and hyperoside (r = −0.96 *p* = 0.03) show negative correlation in 72 h incubation. In HeLa cells, *p*-hydroxybenzoic acid (r = −0.99 *p* = 0.01), hesperidin (r = −0.99 *p* = 0.004) in 24 h incubation and *p*-hydroxybenzoic acid (r = −0.99 *p* = 0.05) in 48 h incubation show negative correlation. Ferulic acid showed a positive correlation (r = 0.99 *p* = 0.04) and *p*-hydroxybenzoic acid showed a negative correlation (r = −0.99 *p* = 0.05) in A549 cells after 48h of incubation period. There is a statistically significant correlation between the change in the amount of major compounds of *A. gymnolobus* methanol extract and the IC_50_ values obtained in A549, HeLa and MDA-MB-231 cells (Figure 3, Appendix A). Hesperidin, the most abundant component, shows a strong negative correlation in cytotoxicity in HeLa cells at 24 h incubation. Catechin, one of the other major components, shows a strong negative correlation with cytotoxicity in MDA-MB-231 cells in 72 h incubation. This suggests that there may be a reason for the very low IC_50_ values in HeLa and MDA-MB-231 cells.

### 2.9. Docking Studies

Binding energy and poses of compounds to B-cell lymphoma 2 (BCL-2), Cyclin dependent kinase 1 (CDK1), Tumor necrosis factor alpha (TNFα), Histone deacetylase 2 (HDAC2), and Penicillin binding protein 2a (PBP2a) were calculated with Autodock Vina, and the results are given in Table 6. According to the docking scores, hesperidin showed the best binding affinity with proteins among the compounds. Table 6 shows that apigenin 7-glucoside, hyperoside, and catechin also have remarkable binding affinities with various proteins. In addition, interaction profiles of hesperidin with target proteins and 2D and 3D illustrations of these interactions are shown in Figure 4.

According to Figure 4, the interacting residues in the BCL-2-Hesperidin complex showed the binding energy of −9.3 kcal/mol, so these interactions are defined as one carbon-hydrogen bond with Gly83 and three hydrogen bonds with Glu74, Asp41, Ala87, Ala38, Tyr140, and Phe42 (Figure 4A). The CDK1-Hesperidin complex included six hydrogen bonds (Asp86, Asp146, Glu12, Ser84, Leu83), and two carbon-hydrogen bonds (Lys33, Gln132) with the binding energy of −8.7 kcal/mol (Figure 4B). On the other hand, the HDAC2-Hesperidin complex has 11 non-covalent interactions with the binding energy of −7.7 kcal/mol and hydrogen bonds that four-amino acids, namely Asn89, Pro200, Tyr198, and Tyr297, two carbon-hydrogen bonds with His 135 and Phe199 (Figure 4C). The binding poses of the TNFα-Hesperidin complex with the lowest binding energy (−9.0 kcal/mol) showed seven hydrogen bonds with Ser60, Lys98, Leu120, Gly121, and Tyr151, one carbon-hydrogen bond with Lys98, and one pi-donor hydrogen bond with Tyr119. Finally, the interactions of PBP2a with hesperidin, which was selected as an antimicrobial target, showed a low binding energy of −9.4 kcal/mol. Upon analysis, the PBP2a-Hesperidin complex’s interaction model revealed the presence of nine hydrogen bonds and one carbon-hydrogen bond.

BCL-2, CDK1, HDAC2, TNF, and PBP2a co-crystallized complexes with binding energies of −11.6 kcal/mol, −9.1 kcal/mol, −7 kcal/mol, −8.9 kcal/mol, and −7.5 kcal/mol, respectively, presented RMSD values of 1.47 Å, 1.48 Å, 0.84 Å, 0.95 Å, and 1.77 Å as a result of redocking experiments for the validation.

## 3. Discussion

Cancer is one of the most important health problems threatening human life worldwide [22,23]. Annual cancer-related deaths are estimated to increase to over 11 million by 2030 [24]. Currently chemotherapy and radiotherapy are the most frequently practiced treatment options for cancer treatment, however potential adverse/side effects in patients significantly affect treatment success. Therefore, investigations for discovery of safer treatment options are continuously carried out by worldwide scientists to reach the supreme target. Among the possible sources for such investigations, plants are considered to be inexhaustible resources in developing agents for cancer treatment. More than 60% of the chemotherapeutic agents used in cancer treatment are natural products or their full/semi-synthetic derivatives [25].

*Astragalus* plants have been shown to possess a wide range of biological properties, including antimicrobial, antifungal, antioxidant, cytotoxic, leishmanicidal, immunomodulatory, antidiabetic, and anti-inflammatory properties. At least 200 active ingredients were identified by phytochemical analyses, mostly as saponins of the cycloartane and oleanane types, flavonoids in their free or glycosidic forms, sterols, and polysaccharides [9,26].

One of the largest classes of secondary metabolites that have been extensively studied in literature are phenolic compounds, which are ubiquitous in all types of plants [27]. In vitro studies have shown that phenolic compounds have strong antioxidant and anti-cancer properties [28,29]. The abundance of phenolic chemicals and flavonoids found in the *Astragalus* genus is well known. Total phenolic content of methanol extracts obtained from *A. gymnolobus* plant was investigated. The highest phenolic content was detected in the Lm extract (374.41 ± 14.37 mg GAE/g). According to the amount of phenolic content, from more to less, the extracts can be sorted as follows: Lm, WPm, Rm, Sm and Fm.

Flavonoids are polyphenols containing broad heterogeneous groups consisting of benzo-*p*-pyron derivatives, which are mostly found in fruit, vegetable, hazelnut, peanut, red wine, tea and the parts of medicinal and aromatic plants [30,31,32,33]. Flavonoids have the ability to modulate cell differentiation and cell cycle [34,35] and can inhibit the enzymes in the oxidation system such as 5-lipoxygenase, cyclooxygenase, monoxygenase, xanthine oxidase and also the growth of arteriosclerotic plaques by preventing lipid peroxidation. In addition, flavonoids have been shown to have antioxidant, antimicrobial, antiviral, anti-inflammatory, antiproliferatie and proapoptotic effects [31]. The extracts can be ranked from the highest to the lowest flavonoid content as follows: Lm, WPm, Rm, Sm and Fm.

In addition to the total phenolic and flavonoid assays, amounts of thirty different standard phytochemicals were also determined by using LC–ESI–MS/MS analysis and the results were given in Table 1. The data obtained clearly showed that none of the extracts contained pyrocatechol, 3,4-dihydroxyphenylacetic acid, caffeic acid, verbascoside, taxifolin, sinapic acid, luteolin 7-glucoside, rosmarinic acid, 2-hydroxycinnamic acid, pinoresinol, eriodictyol, luteolin and apigenin. Overall, the findings from this section seem consistent with the extracts’ total phenolic/flavonoid concentrations. As far as our literature survey could ascertain, in the study conducted by Sarikurkcu et al. (2020) on the phytochemical content of *A. gymnolobus*, it was reported that the methanol extract is rich in hesperidin, hyperoside, (+)-catechin, vanillic acid, protocatechuic acid and *p*-hydroxybenzoic acid [9]. The results from this study appear to be quite similar to the phytochemical composition given in the present study. In another study which methanol extracts from field- and in vitro grown *A. gymnolobus* leaves were investigated for the presence of a limited number of phenolic compounds such as coumarin, apigenin, caffeic acid, rutin hydrate, quercetin, luteolin-7-O-β-D glucoside and myricetin, rutin was identified as the main component in the extracts [20]. In studies on other *Astragalus* species, Haşimi et al. (2017) showed that varying amounts of hesperidin and hyperoside were the most abundant flavonoids in *Astragalus* species in an attempt to detect twenty-four phenolic compounds and three non-phenolic organic acids in three *Astragalus* species [36]. In addition, hyperoside, apigenin, *p*-coumaric and ferulic acids were detected as the main compounds in parts of *A. macrocephalus* Willd. subsp. *finitimus* (Bunge) D.F.Chamb [37]. In another study, seven phytochemicals (syringe acid, p-coumaric acid, o-coumaric acid, luteolin, ferulic acid, hesperidin and benzoic acid) were detected in methanolic extracts from different parts of *A. ponticus* Pall. [38].

As can be seen from Table 4, Lm extract is richer than the others in terms of both flavonoids and phenolics. This circumstance is also supported by the results of the antioxidant activity tests. Total antioxidant capacity (TOAC) and antioxidant capacity equivalent to Trolox (TEAC) of Lm extract were found to be generally higher than the others. Our data suggest that the antioxidant properties of the extract may be related to the amount of phenolic and flavonoid content. The reducing powers of the extracts were determined by using CUPRAC and FRAP test systems. Although the activity potentials of WPm and Lm extracts were close to each other, Lm extract showed the highest activity in both tests. Sarıkurkcu et al. (2020) reported that among tested *Astragalus* species, *A. gymnolobus* methanolic extract showed the highest activity in both CUPRAC and FRAP tests [9]. All extracts demonstrated a concentration-dependent free radical scavenging activity (%) by scavenging DPPH radical. WPm and Lm, with IC_50_ values of 143 ± 7.42 and 152 ± 6.13 μg/mL, respectively, exhibited superior radical scavenging activity among the tested extracts.

The extract concentration at which 50% of the cells are prevented from growing is shown by the IC_50_ values. According to the National Cancer Institute (NCI) standard of cytotoxicity, a crude extract is classified as active, moderately active, or inactive if its IC_50_ values are less than 20 μg/mL, between 20 and 100 μg/mL, or more than 100 μg/mL, respectively [39]. Methanol extracts had the highest cytotoxic effect on HeLa among all cells. Except the stem, all extracts exhibited strong cytotoxic effects on HeLa cells. Lm (IC_50_ = 0.069 μg/mL) induced strongest cytotoxicity in HeLa cells and can be classified as an active crude extract according to NCI. Except the WPm, other extracts exhibited strong cytotoxic effects on MDA-MB-231 breast cancer cells. Sm (IC_50_ = 7.121 μg/mL) induced strongest cytotoxicity in MDA-MB-231 cells and can be classified as an active crude extract according to NCI. Unfortunately, the extracts did not exhibit strong cytotoxic effects on lung cells and can be classified as inactive according to NCI. The anti-cancer potential of any tested compound is increased in proportion to its ability to effectively and selectively exhibit maximum efficiency in inhibiting cancer cell growth and/or proliferation, and by not producing toxic responses in normal cells. Indeed, determination of the cytotoxic activity of the tested extracts only in cancer cells is not sufficient to claim that the agent exhibits antiproliferative or anticancer effects. In our study, the selective cytotoxic responses of extracts of different parts of *A. gymnolobus* plant were shown on HeLa and MDA-MB-231 cells, in other words, the extracts at effective doses and incubation times did not produce any cytotoxic effects on 293T renal epithelial cells that we used as non-cancerous cells.

Flavonoids are known to have potent proapoptotic and hence anticancer effects. We speculate that this cytotoxic effect is caused by the flavonoid and phenolic compounds that we found to be high in Lm extract. The main compounds in the extracts exhibited significant biological activities in earlier studies. For example, hyperoside is a main compound in the genus *Hypericum* L. and this compound exhibits promising biological abilities [40,41,42,43]. Additionally, similar properties were also reported for hyperoside [44], *p*-hydroxybenzoic acid [45,46] and (+)-catechin [47,48] *p*-coumaric acid [49]. The obtained findings suggest a substantial correlation between the stated cytotoxicity and changes in the quantity of hesperidin in HeLa cells after 48 h of incubation. Hesperidin suppresses the growth of HeLa cells via blocking endoplasmic reticulum stress pathways and cell cycle arrest-mediated death [50]. Hesperidin and its derivatives were found to have anticancer effects in HepG2 hepatocarcinoma and HeLa cells [51]. From this vantage point, it is possible to correlate the biological activity of methanolic extracts from *A. gymnolobus* with the existence and amount of these chemicals. These findings are consistent with the theory that both hesperidin and hyperoside considerably enhance the biological activity of certain *A. gymnolobus* components.

Activating the mechanism of apoptosis in tumor cells plays an important role in the development and progression of cancer. Induction of apoptosis in tumor cells is recognized as a therapeutic strategy in cancer treatments. In this study, we investigated whether the extracts at cytotoxic doses at IC_50_ values induced the apoptosis in the cells by detecting the presence of apoptotic DNA fragments in the cytoplasm of cancer cells. The increase in absorbance values of the extracts depending on the dose and time is one of the important findings that makes us claim that the mechanism of death in these cancer cells is apoptotic. Apoptosis can be triggered by both internal (mitochondrial) and external (death receptor) pathways. Caspases as cysteine proteases are the most important determinants of the apoptotic mechanism and are divided into three basic classes in mammalian cells; initiator caspases, effector caspases and inflammatory caspases. The absolutely activated Caspase-3 in the apoptotic mechanism stimulated by the intrinsic or extrinsic pathway is a protein that plays a key role in the apoptotic mechanism because it cleaves the cellular proteins and induces irreversible death of the cell [52,53]. In our study, we investigated whether the extracts that we determined to be cytotoxic and trigger DNA fragmentation, change the activity of Caspase-2, -3, -6, -8 and -9 by ELISA. All extracts led to an increase in the activity of the Caspase-3 protein, confirming that the mechanism of death is likely to be caspase-dependent apoptosis. Increased Caspase-2 or Caspase-8 activity by some extracts has been noted as the first indication that these extracts could induce apoptosis by using the death receptor pathways. However, these data should be supported by investigating different protein expression and activities involved in the external pathway. At this point, we argue that the extracts induce the mechanism of apoptosis by triggering DNA fragmentation following Caspase-3 activation.

Molecular docking research is an advantageous strategy that streamlines the execution of in vitro and ex vivo studies through the elimination of high-throughput screening of a vast array of compounds typically employed in the drug discovery pipeline [54]. Therefore, molecular docking experiments have the capability to offer insight into the relative potency of compounds present within the *Astragalus* methanol extract. Based on Pearson correlation analysis of the change in amounts of compounds derived from *A. gymnolobus* methanol extract compared to IC_50_ extract values, the products that were chosen for molecular docking have been determined. A comprehensive analysis of extant literature was undertaken to ascertain the potential protein targets of these compounds [54,55,56,57,58,59,60]. Hesperidin has the highest scores in comparison to other compounds, based on docking results obtained with Autodock Vina. The computed binding energy values were lowest for the protein targets BCL-2, CDK1, HDAC2, TNFα, and PBP2a, exhibiting energy of −9.3 kcal/mol, −8.7 kcal/mol, −7.7 kcal/mol, −9.0 kcal/mol, and −9.4 kcal/mol, respectively. A study conducted by Taghizadeh et al. (2022) examined the molecular interactions of hesperidin, a natural flavonoid extracted from *Citrus limetta*, with specific proteins, including BCL-2, BCL-W, MCL-1, and ERα. The findings indicated that the Hesperidin-BCL-2 complex showed a very low binding energy of −8.0 kcal/mol [57]. On the other hand, in a study by Boyenle et al. (2022) fifty-five plant-derived bioactive compounds with currently detailed anti-inflammatory activities were screened for their affinity for TNF-alpha employing a molecular docking strategy utilizing three different program packages (iGEMDOCK, MOE and SAMSON). According to the results, hesperidin was ranked fifty-fourth in iGEMDOCK, fifty-third in MOE, and fifty-fifth in SAMSON. Even though different programs than Autodock Vina were utilized within the consideration, the results show that hesperidin performed well, comparable to our study [60].

All antimicrobial activity results indicated that the methanol extract of *A. gymnolobus* had low or no antimicrobial activity against the tested bacteria. While antimicrobial activity was observed against *S. pyogenes* and *S. aureus* strains in the disk diffusion test, the activity against only *S. pyogenes* was detected in the broth dilution test. According to these results, *S. pyogenes* was the most sensitive bacterium to the extracts. Similarly, Turker and Koyluoglu (2012) evaluated the biological activities of 3 different extracts (aqueous, methanol and ethanol) from 8 different Turkish endemic plants and found that the methanol extract of *A. gymnolobus* was effective only against *S. pyogenes* with a 7 mm inhibition zone [21]. In another study evaluating the antibacterial activities of methanol extracts of in vitro-regenerated and field-grown *A. gymnolobus*, field-grown leaves exhibited antibacterial potential against only *S. pyogenes*, *S. aureus*, and *S. epidermidis*. The highest inhibitory activity was obtained against *S. pyogenes* (11.0 ± 0.3 mm inhibition zone) [20]. In the literature, there are other studies investigating the antibacterial activity of different *Astragalus* species on the bacterial strains that we used in our study. Türker and Yıldırım (2013) found that all extracts (aqueous, methanol, ethanol) of *A. brachypterus* had activity against only *S. pyogenes* [61]. Keskin et al. (2018) also showed that the methanol extract of *A. gymnalopecias* Rech.f. was effective only on *S. pyogenes* (12 mm inhibition zone) [62]. Gram-negative bacteria have a cell wall that consists of three layers. The first layer is the outer membrane (OM), which differentiates Gram-negative bacteria from Gram-positive bacteria and makes them more protective and resistant. Thus, Gram-negative bacteria are known to be more resistant to antibiotics than Gram-positive ones [63]. Probably due to these properties of Gram-negative bacteria, we could not observe any activity of the extracts on Gram-negative bacteria tested in this study. Differently, there are studies showing that *Astragalus* extracts are effective against Gram-negative bacteria as well as Gram-positive bacteria. *Astragalus* sp. methanol extract (5 mg) was reported to show activity against *S. aureus* (20 mm) and *E*. *coli* (15 mm) [64]. The methanol extract of *A. pelecinus* (L.) Barneby was found to exhibit antimicrobial activity against *S. aureus*, MRSA and *E. coli* with 51.5 mg/mL MIC value [65]. In the study investigating the antimicrobial activity of the methanol extracts obtained from underground and aerial parts of *A. argaeus* by Albayrak and Kaya (2019), it was determined that the extract only showed activity against *P. aeruginosa* in the inhibition zone range of 7–8 mm [66]. Teyeb et al. (2012) found that the methanol extract of *A. gombiformis* Pomel showed activity against *P. aeruginosa* and *S. typhimurium* with an inhibition diameter of 14 and 13 mm, respectively [67]. Furthermore, this result appears to be higher than the activity against *S. aureus* (8 mm) and *S. epidermidis* (11 mm). Unlike our results, there are also studies reporting that *Astragalus* species have no effect against the bacteria we tested. It was determined that methanol extracts of *A. christianus* Sm., *A. campylosema* Boiss., *A. lineatus* Lam. and *A. globosus* Lam. species were not effective against any tested bacteria [68]. Adiguzel et al. (2009) found that the methanol extracts obtained from the tested *Astragalus* sp. did not have antimicrobial activity on the microorganisms [69].

## 4. Materials and Methods

### 4.1. Plant Material

*A. gymnolobus* was collected from Korkuteli district, Antalya province in Turkey during the flowering period in May 2011, 38°29′56” N, 31°18′49” E. The taxonomic identification of the species was carried out by Dr. R. Süleyman Gokturk (Faculty of Science, Akdeniz University, Turkey), where a voucher specimen has been deposited (Herbarium number: 5670). The collected material dried for a week at ambient temperature and then ground finely.

### 4.2. Extraction

The dried whole plant and the separated plant parts i.e., flowers, roots, stems, leaves were individually weighed 50 g each and extracted with methanol. Then each extract was filtrated and concentrated to dryness under reduced pressure with rotary evaporator at 40 °C. The dried extracts were stored in the dark at 4 °C until further processing.

### 4.3. Identification of Phenolic Compounds by Liquid Chromatography–Electrospray Tandem Mass Spectrometry (LC–ESI–MS/MS)

The phytochemical compositions of the extracts were analyzed quantitatively. Quantitative analyzes were carried out by using LC–ESI–MS/MS technique (Agilent Technologies 1260 Infinity liquid chromatography system, Santa Clara, CA, USA) [70]. Information on the details of the performed analyzes are included in the Appendix A. All tests were carried out in triplicate. In order to determine the degree of statistical difference, Tukey’s test was used.

### 4.4. Cell Culture

The human breast adenocarcinoma (MDA-MB-231, ATCC^®^ HTB-26™), human cervix adenocarcinoma (HeLa, ATCC^®^ CCL-2™), human lung carcinoma (A549, ATCC^®^ CCL-185™) and kidney epithelial cell line (293T, CRL-1573) were purchased from the American Type Culture Collection (ATCC) (Rockville, MD, USA). The cells were maintained in RPMI 1640 medium supplemented with 10% fetal bovine serum (FBS), 2 mM L-glutamine, 1 mM sodium pyruvate and 0.02 mM non-essential amino acids. The cells were maintained at 37 °C in a humidified atmosphere of 5% CO_2_.

### 4.5. Cell Proliferation (WST-1) Assay

Cell proliferation was studied by using WST-1 cell proliferation kit (Roche, Kat. No: 11 644 807 001). The assay is based on the cleavage of the tetrazolium salt to formazan by cellular mitochondrial dehydrogenase [71]. The cells were seeded at 1 × 10^4^ cells per well in 100 µL complete medium onto 96-well plates. Cells were allowed to attach the plates for 24 h. After the cells reaching to 80–90% confluence, the medium was removed, and the cells were treated with various concentrations of extracts prepared in 1% FBS containing complete medium. Immediately after the treatment, cell viability in a single column was determined and recorded as time zero. The plain medium with 1% FBS was used as negative control. Each treatment was performed in eight well replicates. The cells were grown at 37 °C for time course (24, 48 and 72 h). At the end of the incubation, the medium was gently aspirated to terminate the experiment, the WST reagent was added and further incubated under the same conditions for 4 h. Then the kit protocol was followed. The absorbances at 450 nm were measured in a microplate reader Thermo Labsystem Multiscan Spectrum, Thermolabsystem, Chantilly, VA, USA), using wells without cells as background. The sample readings calculated by subtracting the average of background absorbances. The half-maximal inhibitory concentration (IC_50_) of each extract was derived by a nonlinear regression model (curve-fit) based on sigmoidal dose response curve (variable slope) and computed using Graph-Pad Prism, version 4.00 (Graph-Pad Software, San Diego, CA). The growth inhibition was determined using: Growth inhibition (%) = [(mean OD value of control group − mean OD value of treatment group)/mean OD value of control group] × 100%.

### 4.6. DNA Fragmentation

The measurement of cytoplasmic histone-associated DNA fragments (mono- and oligonucleosides) after induction of cell death by extracts was performed with the Cellular DNA Fragmentation ELISA kit (Roche Diagnostics, 11 644 807 001). After treatment, cells were lysed with the incubation buffer as described by the manufacturer and the cytoplasmic fraction recovered. ELISA was performed according to the manufacturer’s protocol [72].

### 4.7. Measurement of Caspases Activities

Caspase’s activities were assessed in cytosolic fractions using a caspase colorimetric protease assay kit that included substrates for Caspase-2, -3, -6, -8, -9 (Apotarget kit, Cat. No. KHZ 1001, Invitrogen Corporation, Waltham, MA, USA) according to the protocol provided by the manufacturer [73]. The enzymatic activities of caspases in the cell lysates are directly proportional to the color reaction. Comparison of the absorbance of apoptotic sample with the control allows determination of the fold increase in caspase-activity.

### 4.8. Antimicrobial Activity

#### 4.8.1. Bacteria

Antimicrobial activities of the methanolic extracts were tested individually against a range of 16 ATCC standard bacteria strains which are clinically important Gram (+) and Gram (-) pathogens. The bacteria tested and antibiotics used as positive controls are listed in Table 3. Bacterial strains were cultured overnight at 37 °C on Blood Agar (Becton Dickinson, Franklin Lakes, NJ, USA) before testing.

#### 4.8.2. Disc Diffusion Method

The standard disc diffusion method which has been recommended by CLSI (Clinical and Laboratory Standards Institute) [74] was employed for determination of the antimicrobial activity of the methanolic extracts. Suspensions of the tested bacteria (1 × 10^8^ cells per ml) were spread on Mueller Hinton Agar (BD Diagnostics, Heidelberg, Germany). The extracts were sterilized by filtering through a 0.22 µm filter and sterile paper discs (6 mm diameter) were impregnated with 20 µL of the extracts. Standard antibiotic discs, recommended by CLSI, which were suitable for microorganisms, were placed into the same plates as positive controls and methanol without the extract was used as a negative control. The discs were deposited on the surface of inoculated agar plate. These plates were incubated at 37 °C for 24 h. The diameters of the inhibition zones were calculated in millimeters. Each assay was performed quadruple.

#### 4.8.3. Broth Microdilution Method

Broth microdilution method was used to detect MIC values of the extracts [75]. The extracts (16,384 μg/mL) dissolved in Mueller Hinton Broth (MHB, Merck KGaA, Darmstadt, Germany). Then, geometric dilutions ranging from 1 to 8192 μg/mL of the extract were prepared in a 96-well microtiter plate. Bacteria strains were suspended in cation adjusted-MHB at a final density of 5 × 10^5^ cfu/mL and each well was inoculated with tested bacteria strain. The growth control (medium + bacteria) and the sterility control (only medium) were studied in each microdilution plate. The same procedure was also applied to each control antibiotic. Microdilution plates were incubated at 35 ± 2 °C for 18–24 h. The MIC values were determined by comparing the growth density in the wells containing antibiotics with those in the control wells used in each test set. Each assay was performed quadruple.

### 4.9. Antioxidant Activity

#### 4.9.1. Total Phenolic Contents of the Extracts

Total phenolic contents of *Astragalus* extracts were determined by using the Folin-Ciocalteu reagent according to the method of Singleton and Rossi using gallic acid as standard [76]. In this method, Folin-Ciocalteu reagent was used to form blue molybden-tungsten complex with the phenolic compounds in the extracts. Total phenol contents were expressed μg gallic acid equivalents per mg of the extracts.

#### 4.9.2. Determination of Total Flavonoid Contents

Total flavonoid content was measured by aluminum chloride colorimetric method [77]. Total flavonoid content was expressed as quercetin equivalents (mg RE/g).

#### 4.9.3. DPPH Radical Scavenging Assay

The free radical scavenging activity of *Astragalus* extracts was determined by the DPPH method. DPPH (2,2-diphenyl-1-picrilhydrazil) is a stable free radical used to determine the free radical scavenging effect of natural compounds. DPPH is reduced to yellow diphenyl picryl hydrazine in the presence of a hydrogen donor antioxidant compound. The absorbance of the resulting yellow compound was measured colorimetically [78]. The percentage inhibition was calculated as follows; percentage inhibition = [(A_0_ − A_1_)/A_0_] × 100, where A_0_ is the absorbance of the blank sample and A_1_ is the absorbance of the sample.

#### 4.9.4. Superoxide Radical (•O_2_^−^) Scavenging Assay

Superoxide is the most produced radical in the body and is also effective in the formation of other radicals. Superoxide can be produced in the presence of xanthine and xanthine oxidase in vitro. Superoxide radical (•O_2_^−^) scavenging activity was measured by the method of Aydın et al. (2001) [79]. Ascorbic acid was used as positive control.

#### 4.9.5. Determination of Copper Ion Reducing Effect (CUPRAC)

The copper ion reductive effect of the extracts was evaluated using the method based on the reduction of Cu^2+^ to Cu^1+^ in the presence of neocuproin and subsequent complexation with neocuproin [80].

#### 4.9.6. Determination of Fe^2+^ Chelating Effect

Fe^2+^ ions complex with ferrozine to form a dark purple color. The formation of this color is prevented in the presence of any metal chelating compound [81].

#### 4.9.7. Determination of Lipid Peroxidation Inhibitory Effect in β-Carotene/Linoleic Acid System

This method is based on the discoloration of β-carotene due to the radicals formed by linoleic acid in an emulsion. In the presence of an antioxidant, the discoloration is inhibited and the reaction is measured spectrophotometrically [82].

#### 4.9.8. Determination of Total Antioxidant Capacity

This method is based on the reduction of Mo ^6+^ to Mo ^5+^, resulting in the formation of a green phosphate/Mo ^5+^ complex [83]. The results were expressed as ascorbic acid equivalents.

#### 4.9.9. Determination of Antioxidant Capacity Equivalent to Trolox

The ABTS radical is a cationic free radical formed by the reaction of 2,2′-azino-bis (3-ethylbenzothiazoline-6-sulphonic acid) (ABTS) with potassium persulfate. The dark green color of the ABTS radical disappears in the presence of a hydrogen donor antioxidant. Results are calculated as equivalent to Trolox in water [84].

### 4.10. Pearson Correlation Analysis

The relationship between the changes in the amounts of phytochemicals in the methanol extract of *A. gymnolobus* and the applied IC_50_ doses of the extracts was examined by Pearson correlation analysis using GraphPad Prism 9 software. The heat map was prepared bidirectionally in order to observe the variability between the changes in the amounts of phytochemicals and the IC_50_ doses [85,86].

### 4.11. Docking Studies

Firstly, the ligands to be used for the molecular docking study were selected by considering the correlation between compound analysis and IC_50_ values. Based on the literature, target proteins associated with selected ligands were determined [54,55,56,57,58,59,60]. The crystal structures of BCL-2 (PDB ID: 6O0K), CDK1 (PDB ID: 6GU6), HDAC2 (PDB ID: 4LXZ), TNFα (PDB ID: 2AZ5), and PBP2a (PDB ID: 1MWT) were availed from the Protein Data Bank (PDB). UCSF Chimera (version 1.16, UCSF, San Francisco, CA, USA) was used for the docking preparation of protein structures. The addition of polar hydrogen atoms and AMBER ff14SB partial charges to the 3D structure followed the removal of the original ligand and water. All selected ligands were retrieved from PubChem, and using Chem3D Pro (version 12.0, CambridgeSoft/PerkinElmer, Waltham, MA, USA), the energy minimization of all ligands was performed by the MM2 method. Ligand preparation was completed using the UCSF Chimera. A grid box was generated according to the binding site of the co-crystallized ligand. Possible protein-ligand binding modes were calculated based on the grid box. Finally, molecular docking was done using AutoDock Vina (version 1.2.0, The Scripps Research Institute, San Diego, CA, USA, EEUU) software. Visualization of the results and analysis of protein-ligand interactions were performed with BIOVIA Discovery Studio Visualizer (version 21.1.0.20298, Dassault Systèmes, Vélizy-Villacoublay, France). The ligands that were co-crystallized were extracted and subsequently subjected to re-docking in the active sites to authenticate the outcome of the docking process. To appraise the quality of co-crystallized ligands, their respective Root Mean Square Deviation (RMSD) values were acquired. A threshold of RMSD value lower than 2 Å is deemed to indicate an accurate prediction of the ligand-protein complex conformation through computational methods [87]. The computations were performed using the DockRMSD webserver [88].

### 4.12. Statistical Analysis

The values obtained after each application and each set were checked for compliance with the normal distribution and variation homogeneity. Parametric and non-parametric test methods were used for the appropriate and non-parametric tests, respectively. ANOVA-MANOVA was used in the analysis of the data that meets the parametric hypothesis criteria, otherwise Kruskal-Wallis hypothesis tests were used. In the tables to be prepared for each analysis, probability of statistical significance (reliability level) (*p*), mathematical degree of freedom (df) and calculated test values are given. The statistical error (α) was taken as 0.05 for each test. The results were interpreted separately considering their significance. Hypothesis tests were performed using SPSS for Windows 17.0.0-19.00 (SPSS Inc.; Chicago, IL, USA). 1989–2010 and Microsoft Excel XP. Prepared datasets were recorded on stationary media for later use. Depending on the results obtained from the experiments in this study, the above-mentioned package programs were used during the calculation of the dose-dependent IC_50_ value. In probit-based statistical evaluations; the concentrations that inhibited viability between 25% and 75% in the trials were pre-evaluated. The results of the trials in which the doses capable of providing 50% inhibition can be calculated from these preliminary evaluation results were used in the tables, graphs and evaluations, others were not considered. In the data obtained from the statistical program Instant 3 used for drawing the graphs; *: *p* < 0.05, **: *p* < 0.01 and ***: *p* < 0.001 were used to show the importance. Quantitative analyzes by using LC–ESI–MS/MS technique were carried out in triplicate. In order to determine the degree of statistical difference, Tukey’s test was used.

## 5. Conclusions

As a conclusion, *A. gymnolobus* did not exhibit cytotoxic effects on normal epithelial cells but showed significant cytotoxic effects on cancer cells. This demonstrates that the plant selectively acts on cancer cells. *A. gymnolobus* exhibited significant cytotoxic activities at very low concentrations in HeLa cells. This plant must further be evaluated for cervical cancer, in terms of its anti-cancer properties, and the mechanism of action should be clarified. *A. gymnolobus* significantly increased Caspase-3 activation, causing DNA fragmentation in the cells. Although the type of death caused by the extracts in cancer cells seems to be apoptotic, considering the apoptosis markers used in our study, the mechanism should be studied in more detail. Unfortunately, *A. gymnolobus* extracts did not show the expected antimicrobial activity against the tested bacteria. Hesperidin, hyperoside, *p*-hydroxybenzoic acid, protocatechuic acid and *p*-coumaric acid were identified as main compounds among the extracts. Hesperidin, the most abundant compound in *A. gymnolobus* extract, may be responsible for the cytotoxic effect observed in HeLa cells. Different parts of the plant exhibit important antioxidant properties, so this plant can also be evaluated with its cancer-protective effect. *A. gymnolobus* has the potential to help combat cancer in a double-sided manner in order to prevent cancer with its antioxidant properties and to prevent the growth and/or division of cancer cells by its phenolic content.

## Figures and Tables

**Figure 1 plants-13-00658-f001:**
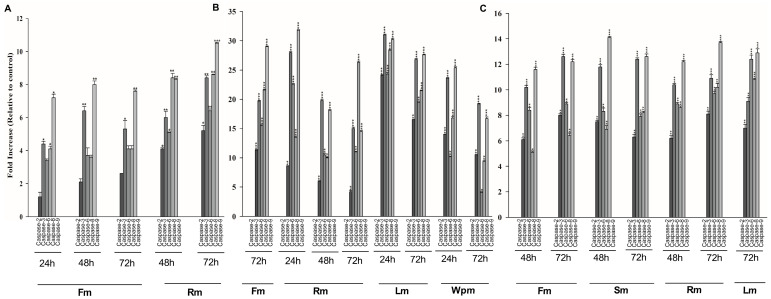
Effect of methanolic extracts on multicaspase activity in cancer cells. (**A**) A549 cells were treated with Fm and Rm extracts at IC_50_ concentration for 24, 48 and 72 h. (**B**) HeLa cells were treated with Fm, Rm, Lm and WPm extracts at IC_50_ concentration for 24, 48 and 72 h. (**C**) MDA-MB-231 cells were treated with Fm, Rm, Sm and Lm extracts at IC_50_ concentration for 24, 48 and 72 h. The statistical analysis of the data was carried out by Student *t*-test. * *p* < 0.05, ** *p* < 0.01 and *** *p* < 0.001 were considered to indicate a statistically significant differences compared to control group.

**Figure 2 plants-13-00658-f002:**
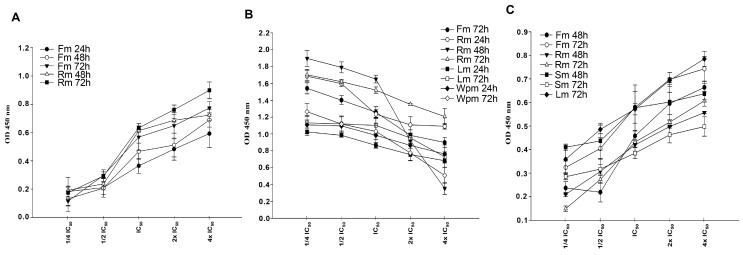
Effect of methanolic extracts on DNA fragmentation of cancer cells. (**A**) A549 cells were treated with Fm and Rm extracts at IC_50_ concentration for 24, 48 and 72 h. (**B**) HeLa cells were treated with Fm, Rm, Lm and WPm extracts at IC_50_ concentration for 24, 48 and 72 h. (**C**) MDA-MB-231 cells were treated with Fm, Rm, Sm and Lm extracts at IC_50_ concentration for 24, 48 and 72 h. The statistical analysis of the data was carried out by Mann-Whitney test and the results were expressed as mean ± SEM.

**Figure 3 plants-13-00658-f003:**
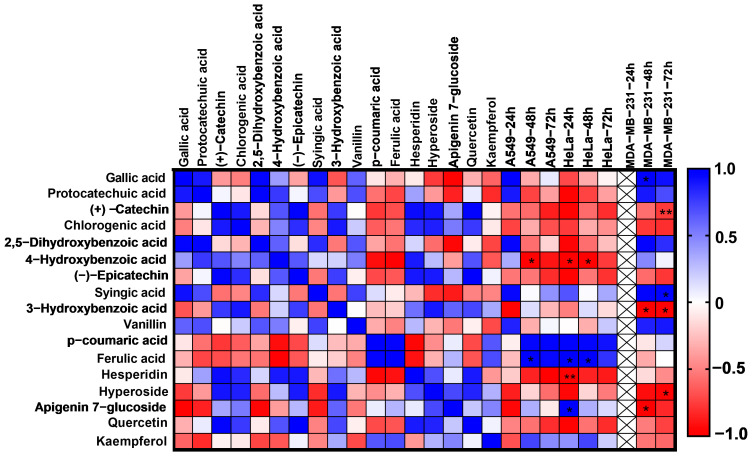
Heatmap analysis of the correlation coefficient matrix between the changes in the amounts of phytochemicals in the *A. gymnolobus* methanol extract and the IC_50_ doses of the extracts applied in A549 (lung cancer cell line), HeLa (cervical cancer cell line) and MDA-MB-231 (breast cancer cell line) by Pearson correlation analysis. The correlation heatmap was prepared bidirectionally. Red and blue color labels indicate positive and negative correlations, respectively. The color label represents the value of the Pearson correlation coefficient, while those with “*” indicate significant values (**, *p* < 0.01; *, *p* < 0.05). Since the IC_50_ dose of the extract could not be determined in MDA-MB-231 cells in 24 h of incubation, it was excluded from the evaluation.

**Figure 4 plants-13-00658-f004:**
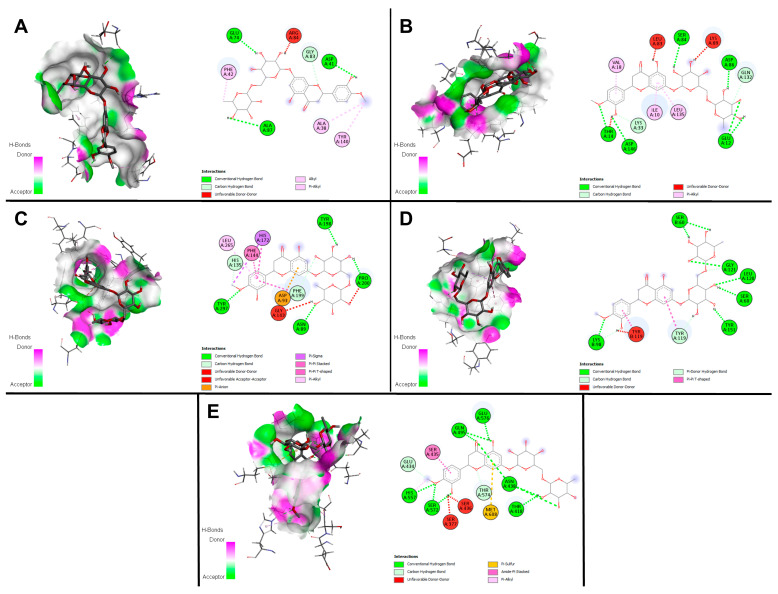
Visualization of the 2D and 3D protein-ligand interaction profile of hesperidin. (**A**) BCL2-Hesperidin, (**B**) CDK1-Hesperidin, (**C**) HDAC2-Hesperidin, (**D**) TNFα-Hesperidin, (**E**) PBP2α-Hesperidin.

**Table 1 plants-13-00658-t001:** Concentration (µg/g extract) of selected phytochemicals in the methanol extracts from different parts of *A. gymnolobus.*

Compounds	Fm	WPm	Rm	Sm	Lm
Gallic acid	39.5 ± 0.6 ^a^	11.4 ± 0.4 ^c^	9.20 ± 0.37 ^d^	14.2 ± 0.1 ^b^	14.0 ± 0.5 ^b^
Protocatechuic acid	348 ± 2 ^a^	20.7 ± 1.1 ^e^	144 ± 1 ^c^	167 ± 3 ^b^	85.2 ± 0.8 ^d^
(+)-Catechin	8.49 ± 0.62 ^c^	nd	365 ± 3 ^b^	454 ± 1 ^a^	12.8 ± 0.6 ^c^
Chlorogenic acid	4.87 ± 0.03 ^d^	8.52 ± 0.53 ^c^	18.8 ± 0.6 ^a^	14.5 ± 0.9 ^b^	3.08 ± 0.15 ^d^
2,5-Dihydroxybenzoic acid	57.5 ± 1.7 ^a^	2.36 ± 0.07 ^d^	11.8 ± 1.2 ^c^	20.4 ± 0.5 ^b^	10.8 ± 0.9 ^c^
*p*-Hydroxybenzoic acid	609 ± 3^b^	66.0 ± 0.8 ^e^	565 ± 3 ^c^	648 ± 1 ^a^	183 ± 1 ^d^
(-)-Epicatechin	nd	nd	9.13 ± 0.46 ^b^	15.5 ± 0.7 ^a^	nd
Syringic acid	16.0 ± 0.8 ^a^	5.52 ± 0.58 ^b^	nd	nd	nd
3-Hydroxybenzoic acid	1.92 ± 0.31 ^a^	2.20 ± 0.10 ^a^	2.26 ± 0.05 ^a^	2.28 ± 0.19 ^a^	1.93 ± 0.08 ^a^
Vanillin	16.0 ± 0.2 ^a^	13.4 ± 1.3 ^ab^	14.5 ± 0.2 ^ab^	12.6 ± 1.5 ^ab^	10.8 ± 0.5 ^b^
*p*-Coumaric acid	57.6 ± 0.6 ^c^	93.1 ± 1.8 ^a^	41.1 ± 0.8 ^e^	48.5 ± 1.1 ^d^	74.3 ± 0.6 ^b^
Ferulic acid	24.9 ± 0.1 ^c^	161 ± 1 ^a^	13.7 ± 0.9 ^d^	15.4 ± 0.2 ^d^	121 ± 2 ^b^
Hesperidin	18581 ± 103 ^b^	147 ± 6 ^d^	40174 ± 136 ^a^	40122 ± 27 ^a^	4854 ± 168 ^c^
Hyperoside	363 ± 4 ^e^	1455 ± 3 ^d^	2444 ± 13 ^b^	2677 ± 13 ^a^	1237 ± 15 ^c^
Apigenin 7-glucoside	1.53 ± 0.18 ^c^	11.1 ± 1.4 ^ab^	13.2 ± 0.3 ^a^	8.99 ± 0.56 ^b^	10.5 ± 0.8 ^ab^
Quercetin	6.07 ± 0.67 ^c^	3.24 ± 0.29 ^d^	34.2 ± 0.4 ^b^	61.8 ± 0.4 ^a^	4.18 ± 0.25 ^d^
Kaempferol	nd	25.5 ± 0.8 ^a^	4.35 ± 0.23 ^c^	19.5 ± 0.1 ^b^	17.1 ± 2.0 ^b^

Within each row, means sharing the different superscripts (^a–e^) show comparison between the samples using Tukey’s test at *p* < 0.05. nd: not detected, Fm: Flower methanol extract, Lm: Leaf methanol extract, Sm: Stem methanol extract, Rm: Root methanol extract, WPm: Whole Plant methanol extract.

**Table 2 plants-13-00658-t002:** IC_50_ values of *A. gymnolobus* Fm, Lm, Sm, Rm and WPm after 24, 48 and 72 h.

IC_50_ Values (µg/mL)					
Extracts	Hours	A549	HeLa	MDA-MB-231	293T
Fm	24 h	2479.518	na	na	na
	48 h	328.782	na	100.284	na
	72 h	409.253	2.950	23.013	na
Rm	24 h	na	1.175	na	na
	48 h	1407.938	2.907	17.742	na
	72 h	966.808	5.031	12.465	na
Sm	24 h	na	na	na	na
	48 h	na	na	7.121	na
	72 h	na	na	10.102	na
Lm	24 h	na	0.069	na	na
	48 h	na	na	na	na
	72 h	na	0.751	7.121	na
WPm	24 h	na	0.940	na	na
	48 h	na	na	na	na
	72 h	na	2.637	na	na

na: not analyzed, Fm: Flower methanol extract, Lm: Leaf methanol extract, Sm: Stem methanol extract, Rm: Root methanol extract, WPm: Whole Plant methanol extract.

**Table 3 plants-13-00658-t003:** Antimicrobial activity results obtained as a result of disc diffusion and test of methanol extracts of *A. gymnolobus.*

Bacteria	Disc Diffusion Results						
	Fm (mm)	Lm (mm)	Sm (mm)	Rm (mm)	WPm (mm)	A (mm)	N (mm)
*Staphylococcus aureus* ATCC 25923	9	-	8	-	9	32 (P)	-
*Staphylococcus aureus* ATCC 29213	8	-	8	-	9	20 (P)	-
*Staphylococcus aureus* ATCC 43300	8	-	8	-	9	15 (FOX)	-
*Staphylococcus epidermidis* ATCC 12228	-	-	-	-	-	20 (VA)	-
*Enterococcus faecalis* ATCC 51299	-	-	-	-	-	15 (VA)	-
*Enterococcus faecalis* ATCC 29212	-	-	-	-	-	20 (VA)	-
*Streptococcus pyogenes* ATCC 19615	10	9	10	9	11	42 (P)	-
*Escherichia coli* ATCC 25922	-	-	-	-	-	21 (AMC)	-
*Escherichia coli* ATCC 35218	-	-	-	-	-	18 (AMC)	-
*Klebsiella pneumoniae* ATCC 13883	-	-	-	-	-	25 (CAZ)	-
*Klebsiella pneumoniae* ATCC 700603	-	-	-	-	-	14 (CAZ)	-
*Enterobacter cloacae* ATCC 23355	-	-	-	-	-	33 (MEM)	-
*Serratia marcescens* ATCC 8100	-	-	-	-	-	30 (MEM)	-
*Proteus vulgaris* ATCC 13315	-	-	-	-	-	35 (FEP)	-
*Salmonella typhimurium* ATCC 14028	-	-	-	-	-	20 (AMP)	-
*Pseudomonas aeruginosa* ATCC 27853	-	-	-	-	-	32 (MEM)	-

Fm: Flower methanol extract, Lm: Leaf methanol extract, Sm: Stem methanol extract, Rm: Root methanol extract, WPm: Whole Plant methanol extract, A: Antibiotic, N: Negative control, P: Penicillin, FOX: Cefoxitin, VA: Vancomycin, CAZ: Ceftazidime, AMC: Amoxicillin/clavulanic acid 2:1, MEM: Meropenem, FEP: Cefepime, AMP: Ampicillin.

**Table 4 plants-13-00658-t004:** Total phenolic (TP) and flavonoid (TF) content, total antioxidant capacity (TOAC) and antioxidant capacity equivalent to Trolox (TEAC) of methanolic extracts.

Extract	TP (mg GAE/g)	TF (mg QE/g)	TOAC (mg AAE/g)	TEAC (µM TE/g)
Rm	200.53 ± 13.62	34.43 ± 4.97	103 ± 1.99	231 ± 13.48
Sm	173.58 ± 5.21	26.57 ± 6.73	86 ± 6.49	187 ± 7.29
Fm	130.27 ± 7.39	9.73 ± 1.44	56 ± 4.37	153 ± 6.38
Lm	374.41 ± 14.37	66.85 ± 3.49	304 ± 7.31	794 ± 8.37
WPm	367.10 ± 15.29	44.78 ± 4.28	258 ± 4.84	678 ± 14.31

GAE: Gallic acid equivalent, QE: Quercetin equivalent, AAE: Ascorbic acid equivalent, TE: Trolox equivalent, Rm: Root methanol, Sm: Stem methanol, Fm: Flower methanol, Lm: Leaf methanol, WPm: Whole Plant methanol.

**Table 5 plants-13-00658-t005:** DPPH radical scavenging effects, Ferric reducing antioxidant power (FRAP), Copper ion reducing effect (CUPRAC) and inhibiting lipid peroxidation effect in β-carotene/linoleic acid system of methanolic extracts.

Extract	DPPH (EC 50 µg/mL)	FRAP (mM FeSO_4_/g)	CUPRAC (mg AAE/g)	β-Caroten (% Activity of 1 mg/mL)
Rm	465 ± 16.34	1.62 ± 0.05	47 ± 2.79	68.41 ± 4.17
Sm	676 ± 12.39	0.91 ± 0.09	31 ± 5.48	59.52 ± 1.34
Fm	759 ± 9.37	0.65 ± 0.03	19 ± 1.65	41.94 ± 2.25
Lm	152 ± 6.13	3.72 ± 0.09	148 ± 4.63	90.37 ± 1.84
WPm	143 ± 7.42	3.15 ± 0.08	132 ± 8.7	83.02 ± 1.94

AAE: Ascorbic acid equivalent, Rm: Root methanol, Sm: Stem methanol, Fm: Flower methanol, Lm: Leaf methanol, WPm: Whole Plant methanol.

**Table 6 plants-13-00658-t006:** The binding energy of the selected compounds (kcal/mol).

Compound	Binding Affinity				
	BCL-2	CDK1	HDAC2	TNFα	PBP2a
Hesperidin	−9.3	−8.7	−7.7	−9.0	−9.4
Apigenin 7-glucoside	−7.7	−7.8	−7.6	−7.9	−9.2
Hyperoside	−7.2	−7.8	−6.5	−8.0	−7.7
Catechin	−6.8	−8.0	−7.5	−7.1	−8.2
Ferulic acid	−5.6	−6.7	−7.6	−5.9	−6.6
3-Hydroxybenzoic acid	−5.6	−4.1	−6.5	−5.4	−5.8
*p*-Hydroxybenzoic acid	−5.2	−5.7	−6.4	−5.2	−5.7
Gallic acid	−5.2	−5.9	−5.9	−5.6	−6.3
Syringic acid	−5.3	−6.2	−5.5	−5.8	−6.0
Co-crystallized ligands	−11.6 ^a^	−9.1 ^b^	−7.0 ^c^	−8.9 ^d^	−7.5 ^e^

^a^: Venetoclax, ^b^: Dinaciclib, ^c^: Vorinostat, ^d^: SPD304, ^e^: Open form-penicillin G.

## Data Availability

The data presented in this study are available in article.

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
