# Peer review of "Phenolic Compound Profiles, Cytotoxic, Antioxidant, Antimicrobial Potentials and Molecular Docking Studies of Astragalus gymnolobus Methanolic Extracts"

_plants, 2024, doi:10.3390/plants13050658_

Round 1

Reviewer 1 Report

Comments and Suggestions for Authors

A brief summary 

The manuscript “Phenolic Compound Profiles, Cytotoxic, Antioxidant, Antimicrobial Potentials and Molecular Docking Studies of Astragalus gymnolobus Methanolic Extracts” reported a phytochemical investigation of Astragalus gymnolobus. The extracts were investigated on multicaspase activity in cancer cells, antimicrobial activity against 16 bacterial strains and antioxidant activity. Hesperidin, the most abundant compound in extract, has the best binding affinity to the targeted proteins.

The references are recent, but some of them should include DOI. Moreover, the references are not correctly reported according to the journal guidelines. Too much self-references are reported.

Conclusions are consistent with the study.

General concept comments

The aim of the work should be explained, and the novelty of the study should be better underlined. In my opinion, reorganizing the introduction with existing literature through sufficient referencing and appropriate coverage of the relevant references could improve readability. The authors need to be specific when making statements based on cited literature. The whole study needs more work: literature support to back-up the data, observation, and interpretation. In my opinion, the manuscript seems written in a careless way and cannot be accepted in the present form. Many points lack a reasonable basis for study.

The abstract at lines 25-26: saying that “the binding could be correlated with the antimicrobial activity” is like a speculation without an enzymatic test or real binding assay. Perhaps, the sentence could be reformulated.

Results:

The 30 compounds identified in the extract are known compounds. Were they first studied? Cite in the introduction the works that report the identification of them.

What does the previous literature say about this type of extracts/activity?

Antimicrobial activity has no sense: we can declare that value of MIC <100 μg/mL for extracts (Gibbons, 2008. Planta Med. 74, 594-602. - Ríos and Recio, 2005. J. Ethnopharmacol. 100, 80-84.) have sense. So such high values, 8192 μg/ml do not provide any information. The table can be removed. Based on what reasoning was it decided to test these 16 bacteria?

Docking study: How were the compounds chosen? who are the crystallised ligands?

Many errors, missing points/spaces and inaccuracies have been reported (ex. Lines 78-162).

Discussion: lines 269-264 should be mentioned in the introduction also.

Hesperidin and hyperoside were found as the major compounds in the reference 33, a study of one of the same author, https://doi.org/10.1016/j.indcrop.2020.112366, which is identical to this. Inserted antimicrobial activity does not exist. So what is the novelty?

Specific comments

Line 20: in “p-coumaric acid, p-nitroaniline” “p” should be in italics. Moreover, “in vitro” “ex vivo” and all Scientific Names should be in italics.

Table 1. It is useless to report compounds nd: not detected.

Figures 1-2-4 are not well visible. The figures are really poor quality!

Table 3 is reported in fuzzy way, and it needs to be better formatting. It would be better to create two separate tables.

Table 6 can be put in supplementary.

Table 7. Binding affinities should report the kcal/mol?

Comments on the Quality of English Language

Minor editing of English language required.

Author Response

You can find the attached point by point response to the comments.

1. Summary

2. Questions for General Evaluation and Reviewer’s Evaluation

Quality of English Language

( ) I am not qualified to assess the quality of English in this paper
( ) English very difficult to understand/incomprehensible
( ) Extensive editing of English language required
( ) Moderate editing of English language required
(x) Minor editing of English language required
( ) English language fine. No issues detected

3. Point-by-point response to Comments and Suggestions for Authors

A brief summary

The manuscript “Phenolic Compound Profiles, Cytotoxic, Antioxidant, Antimicrobial Potentials and Molecular Docking Studies of Astragalus gymnolobus Methanolic Extracts” reported a phytochemical investigation of Astragalus gymnolobus. The extracts were investigated on multicaspase activity in cancer cells, antimicrobial activity against 16 bacterial strains and antioxidant activity. Hesperidin, the most abundant compound in extract, has the best binding affinity to the targeted proteins. The references are recent, but some of them should include DOI. Moreover, the references are not correctly reported according to the journal guidelines. Too much self-references are reported. Conclusions are consistent with the study.

Comments 1: The aim of the work should be explained, and the novelty of the study should be better underlined. In my opinion, reorganizing the introduction with existing literature through sufficient referencing and appropriate coverage of the relevant references could improve readability. The authors need to be specific when making statements based on cited literature. The whole study needs more work: literature support to back-up the data, observation, and interpretation. In my opinion, the manuscript seems written in a careless way and cannot be accepted in the present form. Many points lack a reasonable basis for study.

Response 1: References 4 and 32 are master theses and unfortunately do not have doi numbers. References 20,21,33,36,39,65 and 68 do not have doi numbers and therefore could not be added to the references section.

Comments 2: The abstract at lines 25-26: saying that “the binding could be correlated with the antimicrobial activity” is like a speculation without an enzymatic test or real binding assay. Perhaps, the sentence could be reformulated.

Response 2: This sentence has been completely removed from the abstract.

Comments 3: Results: The 30 compounds identified in the extract are known compounds. Were they first studied? Cite in the introduction the works that report the identification of them. What does the previous literature say about this type of extracts/activity?

Response 3: Studies on the compounds in the extract and their effects were added to the introduction and cited.

Comments 4: Antimicrobial activity has no sense: we can declare that value of MIC <100 μg/mL for extracts (Gibbons, 2008. Planta Med. 74, 594-602. - Ríos and Recio, 2005. J. Ethnopharmacol. 100, 80-84.) have sense. So such high values, 8192 μg/ml do not provide any information. The table can be removed. Based on what reasoning was it decided to test these 16 bacteria?

Response 4: Broth microdulsion test results were removed from Table 3 and the Table was rearranged. These 16 bacteria are among the most clinically important Gram (+) and (-) bacteria. This is briefly described in section 4.8.1 Bacteria.

Comments 5: Docking study: How were the compounds chosen? who are the crystallised ligands?

Response 5: How the compounds studied were chosen is described in the first sentence under 4.11 Docking Studies section as ‘Firstly the ligands to be used for the molecular docking study were selected by considering the correlation between compound analysis and IC50 values’.

Crystallised ligands have been added to Table 6 as a table footer.

Comments 6: Many errors, missing points/spaces and inaccuracies have been reported (ex. Lines 78-162).

Response 6: Errors, missing points/gaps and inaccuracies have been corrected among the text.

Comments 7: Discussion: lines 269-264 should be mentioned in the introduction also.

Response 7: The requested information has been added to the introduction.

Comments 8: Hesperidin and hyperoside were found as the major compounds in the reference 33, a study of one of the same author, https://doi.org/10.1016/j.indcrop.2020.112366, which is identical to this. Inserted antimicrobial activity does not exist. So what is the novelty?

Response 8: The publication https://doi.org/10.1016/j.indcrop.2020.112366 belongs to Cengiz Sarıkurkcu, one of our authors.  In that publication, A. gymnolobus was collected from a different localization (Karamanli, Burdur-Turkey on 20 May 2016 (1150 m., 37° 17′ 24″N 29° 54′ 36″E) at different time. In our study, A. gymnolobus was collected from Korkuteli district, Antalya province in Turkey during the flowering period on May 2011, 38°29’56” N, 31°18’49” E. The study was designed for the possibility of variation in the secondary metabolites of the plant collected from different places and at different times. In the study conducted by Sarıkurkçu, methanol extraction of the whole plant of A.gymnolobus was performed. However, in our study, different parts of the plant were extracted separately and the biological activity of each was investigated and compared with each other. Sarıkurkçu has studied only enzyme inhibitory activity, as biological activity, in the previous study. Antimicrobial or cytotoxic activity was not investigated in that publication.

Comments 9: Specific comments; Line 20: in “p-coumaric acid, p-nitroaniline” “p” should be in italics. Moreover, “in vitro” “ex vivo” and all Scientific Names should be in italics.

Response 9: All text checked, including suggested parts, and all scientific names are italicized.

Comments 10: Table 1. It is useless to report compounds nd: not detected.

Response 10: Compounds that could not be detected were excluded from Table 1.

Comments 11: Figures 1-2-4 are not well visible. The figures are really poor quality!

Response 11: Image quality has been improved for specified figures.

Comments 12: Table 3 is reported in fuzzy way, and it needs to be better formatting. It would be better to create two separate tables.

Response 12: Broth microdilution test results were removed from Table 3 and the Table was rearranged.

Comments 13: Table 6 can be put in supplementary.

Response 13: Table 6 has been transferred to the supplementary file as Table S3.

Comments 14: Table 7. Binding affinities should report the kcal/mol?

Response 14: Since there are instances in the literature (https://doi.org/10.1080/07391102.2022.2124453; https://doi.org/10.3390/molecules26247433), binding affinities are expressed as kcal/mol.

Comments 15: Too much self-references are reported.

Response 15:  Self-references have been removed except the studies on Astragalus species.

Reviewer 2 Report

Comments and Suggestions for Authors

Dear Authors;

some details should be corrected;

The section entitled; Standards and Chemicals should be added. This section should contain listed all of the used standards and chemicals with their purity and supplier.

The compounds nomenclature should be inified; an example; p-coumaric anf 4-hydroxybenzoic acids. It sould be; 4-hydroxycinnamic acid and 4-hydroxybenzoic acid or p-coumaric and p-hydroxybenzoic. Protocatechuic acid is 3,4-dihydroxybenzoic acid. And many similar examples. Please decide about nomenclature of the compounds.

A brief annotation why compounds listed in the lines 88-91 were analyzed (however, not detected) shuld be given; e.g. "Were all of these compounds found in other Astragalus species" (?)

Table 2 ; IC50 values should be given additionally in mM or uM

All of the abbreviation s in the text should be explained in their first appearance.

In the section 2.6 (Antimicrobial activity) should be an information; "Detailed information about bacterial strains tested and antibiotics used is given in the Table 3", as in the experimental part line 512.

Table 7; In the Table thesame points of decimals should be given (e.g. see CDK1 for catechin; is -8, should be -8.0)

Author Response

You can find the attached point by point response to the comments.

1. Summary

2. Point-by-point response to Comments and Suggestions for Authors

A brief summary :Comments and Suggestions for Authors Dear Authors;some details should be corrected;

Comments 1: The section entitled; Standards and Chemicals should be added. This section should contain listed all of the used standards and chemicals with their purity and supplier.

Response1:This is a multidisciplinary study. There is a large amount of different chemicals used in different tests. therefore this section is very long. it can be uploaded as a suplementary file if desired.

Comments 2:The compounds nomenclature should be inified; an example; p-coumaric anf 4-hydroxybenzoic acids. It sould be; 4-hydroxycinnamic acid and 4-hydroxybenzoic acid or p-coumaric and p-hydroxybenzoic. Protocatechuic acid is 3,4-dihydroxybenzoic acid. And many similar examples. Please decide about nomenclature of the compounds.

Response 2: Nomenclature of the compounds are standardized among the text.

Comments 3: A brief annotation why compounds listed in the lines 88-91 were analyzed (however, not detected) shuld be given; e.g. "Were all of these compounds found in other Astragalus species" (?)

Response 3: Compounds that could not be detected were excluded from Table 1.

Comments 4: Table 2 ; IC50 values should be given additionally in mM or uM

Response 4: The plant extract was lyophilized and thawed to W/V. Unfortunately, it is not possible to calculate the molar concentration of the extract as it contains many components.

Comments 5: All of the abbreviation s in the text should be explained in their first appearance.

Response 5: All of the abbreviation s in the text explained in their first appearance.

Comments 6: In the section 2.6 (Antimicrobial activity) should be an information; "Detailed information about bacterial strains tested and antibiotics used is given in the Table 3", as in the experimental part line 512.

Response 6: In the section 2.6 (Antimicrobial activity) should be an information; "Detailed information about bacterial strains tested and antibiotics used is given in the Table 3", as in the experimental part line 512.

Comments 7: Table 7; In the Table the same points of decimals should be given (e.g. see CDK1 for catechin; is -8, should be -8.0)

Response 7: In the Table all points of decimals are corrected

Round 2

Reviewer 1 Report

Comments and Suggestions for Authors

The authors improved the manuscript. A very well work has been done. 

However, I still have some improvement to suggest:

Extraction section should be improved. Authors should report each quantity of the parts in which the whole plant (50 g) has been separated.

details on instrument used LC–ESI–MS/MS should be reported.
